# Molecular Characterization of Citrus Accessions Grown for Pre-Breeding Purposes

**DOI:** 10.3390/cimb47080656

**Published:** 2025-08-14

**Authors:** Israel Felipe Gonçalves Soares, Felipe Cruz Paula, Conceição de Maria Batista Oliveira, José Dias de Souza Neto, Talles de Oliveira Santos, Rafael Nunes de Almeida, Ana Paula Candido Gabriel Berilli, Sávio da Silva Berilli, Taís Cristina Bastos Soares, Jardel Oliveira Santos, Alexandre Cristiano Santos Júnior, Monique Moreira Moulin

**Affiliations:** 1Escola Superior de Agricultura “Luiz de Queiroz”, Universidade de São Paulo (ESALQ_USP), Av. Padua Dias, 11-Agronomia, Piraciacaba 13418-900, SP, Brazil; filipeisraelgoncalves@gmail.com; 2Center for Biosciences and Biotechnology, Universidade Estadual do Norte Fluminense Darcy Ribeiro (UENF), Campos dos Goytacazes 28013-602, RJ, Brazil; felipe.cpaula64@gmail.com; 3Department of Biology, Universidade Federal de Lavras (UFLA), Trevo Rotatório Professor Edmir Sá Santos, Lavras 37200-900, MG, Brazil; oliveiracmb.es@gmail.com; 4Instituto Federal de Ciência e Tecnologia do Espírito Santo, Campus de Alegre (IFES), Rodovia ES-482 Cachoeiro-Alegre, Km 72-Rive, Alegre 29500-000, ES, Brazil; jdiassneto@gmail.com (J.D.d.S.N.); anapaulacg@gmail.com (A.P.C.G.B.); savio.berilli@ifes.edu.br (S.d.S.B.); alexandre.cristiano@ifes.edu.br (A.C.S.J.); mmmoulin@ifes.edu.br (M.M.M.); 5Instituto Capixaba de Pesquisa, Assistência Técnica e Extensão Rural (INCAPER), Centro de Pesquisa e Desenvolvimento em Inovação Norte, BR 101N, km 151, Linhares 29915-140, ES, Brazil; afael.almeida@incaper.es.gov.br; 6Department of Pharmacy and Nutrition, Universidade Federal do Espírito Santo (UFES), Alto Universitario, S/N-Guararema, Alegre 291500-000, ES, Brazil; tais.soares@ufes.br; 7Department of Biology, Universidade Federal do Piauí–UFPI, Campus Ministro Petrônio Portella, Ininga, Teresina 64049-550, PI, Brazil; jardel_santos@ufpi.edu.br

**Keywords:** SSR marker, genetic variability, *Citrus* spp., citrus germplasm

## Abstract

The objective of this work was to analyse the genetic diversity of a population of *Citrus* spp. in the south of the State of Espírito Santo, Brazil, for pre-breeding studies. For that, a total of sixty genotypes were analysed, including ten citrus varieties from four species of the *Citrus* genus. The methodology involved DNA extraction, amplification via polymerase chain reaction, and the use of a set of 16 Simple Sequence Repeat markers. These markers identified 42 alleles, with a variation of one to four alleles per locus, an average heterozygosity value of 0.53, and an average polymorphic information content of up to 0.29 per species. After the analysis, a dissimilarity matrix was generated using Jaccard distance and a dendrogram, revealing the formation of two groups: Group I, comprising *Citrus sinensis* varieties, and Group II, comprising varieties of *Citrus latifolia*, *Citrus aurantifolia*, and *Citrus reticulata*. Our study demonstrated that the combination of these markers allowed for the differentiation of genotypes within the collection. The results obtained are valuable for the future management of the collection and the efficient use of genetic diversity estimation in *Citrus* spp.

## 1. Introduction

The *Citrus* genus belongs to the Rutaceae family and comprises various species, such as *Citrus reticulata* Blanco (mandarins), *Citrus sinensis* L. Osbeck (oranges), *Citrus limon* L., *Citrus latifolia* Tanaka, *Citrus aurantifolia* (C.) Swingle (lemons, limes), *Citrus paradisi* L. (grapefruits), and others [1,2]. These citrus fruits are widely consumed worldwide due to their aromas, flavours, and nutritional value [3]. They are rich in vitamins, fibre, minerals, and phytonutrients essential for human health [4]. Additionally, they play a significant role as a food source and raw material in various industrial sectors, including the production of juices, beverages, and processed foods, and for fresh consumption [5,6].

Citrus fruits are available on global markets, with China being the largest producer (26%), followed by Brazil (13%) [7]. Among the citrus species, *C. sinensis* is the most widely cultivated and has a high profile in Brazil, which is one of the largest producers and exporters of orange juice. In addition, other species such as *C. reticulata*, *C. latifolia*, and *C. aurantifolia* are also economically important, used both for fresh consumption and by industry [8,9]. To ensure the efficiency of the production chain and food safety, detailed knowledge of morphological characteristics is essential for genetic improvement [10,11]. However, the genus *Citrus* shows significant phenotypic variation both within and between species [12].

This phenotypic variation is not always sufficient to accurately estimate the genetic distance between accessions. Therefore, the incorporation of molecular markers serves as an auxiliary tool that provides greater discriminatory power between genotypes [13,14]. Microsatellites, or simple sequence repeats (SSRs), are markers characterised by high levels of polymorphism, codominant inheritance, and multiallelism [15]. SSRs are widely used in diversity studies between populations, including cultivars, due to their simplicity, speed, and efficiency in genotypic and germplasm characterisation in citrus [3,16,17,18].

Germplasm conservation and knowledge are fundamental for citrus cultivation, especially in places where this activity is of great economic and social importance. In the state of Espírito Santo, citrus farming is particularly important in the southern region and in the Caparaó area, where soil and climatic conditions are favourable for citrus cultivation [19,20]. However, although the state is a centre of citrus production, the low productivity of the fruit and the lack of genetic material recommended for the region have contributed to a reduction in the area planted and a decrease in producers’ income. Although producers traditionally maintain genotypes, it is essential to evaluate the genetic diversity of these citrus varieties to ensure their sustainability and continuous improvement. Therefore, the present study aimed to evaluate the genetic diversity of a citrus germplasm collection in southern Espírito Santo using SSR molecular markers.

## 2. Material and Methods

A total of 60 accessions of 10 citrus varieties were evaluated, comprising the following species: I-*Citrus sinensis* (L.) Osbeck, II-*Citrus reticulata Blanco*, III-*Citrus latifolia* and IV-*Citrus aurantifolia* (C.) Swingle. These accessions originated from the germplasm collection of the Federal Institute of Education, Science and Technology of Espírito Santo (IFES)-Campus Alegre, located in the southern region of the State of Espírito Santo, Brazil (latitude 20°45′20′′ S and longitude 41°27′43′′ W) (Figure 1).

Five vigorous leaves of each genotype were collected and preserved in liquid nitrogen. The leaves were then macerated and approximately 300 mg of the resulting plant tissue was transferred to a tube. DNA was then extracted using the CTAB protocol [21]. After removal, the samples were quantified using a spectrophotometer (NanoDrop^®^ ND-1000, ThermoFisher, Wilmington, DE, USA) with a 260/280 nm ratio ≥ 1.8, indicating a high quality of the DNA obtained.

The PCR amplification reactions were performed in a final volume of 15 μL, containing 1.5 μL of 10× PCR buffer, 0.6 μL of MgCl_2_ (250 mM), 4.8 μL of dNTP (2.5 μM), 0.2 μL of Taq DNA polymerase (5 units/μL), 5 μL of genomic DNA (10 ng/μL), 0.75 μL of each oligonucleotide, 1.4 μL of ultrapure water, and 0.2 μL of Taq DNA polymerase (5 units/μL). The thermocycling programme was initiated with a denaturation step at 94 °C for 5 min, followed by 35 cycles of denaturation at 94 °C for 30 s, annealing at 60 °C for 30 s, and extension and 72 °C for 30 s, with a final extension at 72 °C for 5 min [22].

The PCR products were separated by denaturing polyacrylamide gel electrophoresis, at 10% in TBE buffer (1×) (Tris-Boric 45 mM, EDTA 1 mM pH 8.0) for 1 h and 30 min at 100 V, followed by staining with ethidium bromide (25 μL/L) for 30 min, and photographed with BioRad equipment (model Gel Doc XR+ system, Hercules, CA, USA).

Genotyping was performed using 16 SSRs provided by the Laboratory of Biochemistry and Molecular Biology of the Federal University of Espírito Santo Campus Alegre (Table 1) [23,24].

The polymorphic information content (PIC) was calculated based on the 16 SSRs, employing the formula PIC = Σ1 − P2ij, allele frequency, and the expected heterozygosity within the four species. To visualise the genetic diversity relationships, the dissimilarity matrix was calculated based on the Jaccard coefficient, considering the presence and absence of alleles. Clustering was performed using the Unweighted Pair Group Method with Arithmetic Mean (UPGMA), both available in the ‘Genetic Distance Analysis’ routine of the GENES software [25]. The algorithm uses a binary matrix to generate the dissimilarity matrix and subsequently constructs the dendrogram based on the average linkage method. After obtaining the data matrix from the SSR markers, a heatmap analysis with bidimensional hierarchical clustering was performed using the pheatmap package in R version 4.4.2. [26].

## 3. Results and Discussion

A genetic diversity analysis was conducted on 60 citrus varieties, which were submitted to a set of 16 SSRs. The analysis identified 42 alleles, with a variation of one to four alleles between the SSRs evaluated (Table 2). The mean observed heterozygosity was 0.53, suggesting the presence of genetic variation. Seven of the SSRs exhibited higher observed heterozygosity than expected, suggesting a high frequency of heterozygotes in the population under study.

The highest level of heterozygosity was observed in the SSRs CCSMEc2, CCSMEc4, CCSMEc5, CCSMEc7, CCSMEc10, CCSMEc13, and CMS-16; the latter set of markers is the most suitable for the future analysis of species belonging to the genus (Table 2). The availability of these SSR sets, together with others implemented in the citrus microsatellite database, which have polymorphic potential, can be used to infer genetic diversity and population distinction in species of the genus [15].

The SSRs analysed proved to be effective in discriminating diversity between species. The PIC indicated a moderate level of polymorphism, ranging from 0.21 to 0.29 (Figure 2). The *C. sinensis* species had the highest PIC value (0.29), suggesting greater genetic variability within the germplasm (Figure 2). A higher PIC reflects a greater variety of alleles present in the genetic material studied [27]. And the varieties of oranges (*C. sinensis*) exhibited greater dispersion within the group of citrus varieties compared to *C. latifolia*, *C. aurantifolia* e *C. reticulata*, which registered lower values (0.23, 0.21, and 0.21).

Among the markers evaluated, the loci CCSMEc2 and CCSMEc3 were the most informative, while CCSMEc11 and CCSMEc14 showed low efficiency in polymorphism detection. The use of these SSRs was relevant for the study under analysis, as they presented high PICs, given their ability to capture variation within the genetic pool. This, in turn, facilitates a more precise discrimination between different genotypes within a species, and the inclusion of SNPs or InDels can complement low-polymorphism SSRs, providing higher resolution in genetic analyses [17].

Organising this data by species is essential for selecting and recommending the most appropriate markers [15]. Among the SSRs analysed (Figure 2), 13 were informative for *C. sinensis*, with PIC values ranging from (0.21 to 0.50). These values were found to be similar to those previously observed for a sweet orange population, which ranged from 0.37 to 0.43; this finding suggests that there is consistency in the patterns of genetic variability that have been observed [23]. Ten SRRs were informative for *C. latifolia*, and nine were informative for *C. aurantifolia* and *C. reticulata*, showing PIC values (0.30 to 0.38; 0.35 to 0.38) comparable to previous studies in a *Citrus* ssp. population with a PIC between 0.18 and 0.37, supporting marker recommendations based on the data obtained [28].

The UPGMA cluster analysis of the similarity matrix obtained using the 16 SSRs resulted in a dendrogram (Figure 3) with a cophenetic correlation coefficient of 0.96, indicating a well-represented genetic similarity in the structuring of the clusters between the citrus, forming two large groups of similarity. Group I is made up of seven types of orange citrus. *C. sinensis* demonstrates significant genetic diversity, as evidenced by the analysis of agronomic, bromatological, morphological and chemotype characteristics [19,29,30]. However, the investigation revealed minimal genetic variation when employing SSR molecular markers, such as simple sequence repeats (ISSR) [31] and random amplified polymorphic fragments (RAPD) [32].

Oranges have a narrow genetic base among citrus, which can be attributed to their origin from a common lineage, a genetic mutation, or domestication of the *C. sinensis* species [33]. The findings of this study demonstrated that all orange citruses were integrated into a single group within the dendrogram, thereby substantiating their genetic similarity (Figure 3). The formation of clusters for orange citrus can be observed in previous studies involving SSR genetic markers [34,35], confirming the hypothesis that sweet orange cultivars are monophyletic and derived from a single ancestor through mutation and the selection of desirable clones [36,37].

Group II comprises the species *C. latifolia, C. aurantifolia*, and *C. reticulata* (Figure 3). There is a high degree of genetic similarity between the genotypes of each species, suggesting a narrow genetic base or the presence of replications in the germplasm of these genotypes. The genetic similarity between *C. aurantifolia* and *C. latifolia* lemon genotypes demonstrates an intrinsic genetic relationship with other species of the genus. This finding aligns with previous studies that have indicated the classification of both species within the same cluster, owing to the presence of genetic characteristics analogous to those observed in citrus. Consequently, they are regarded as natural hybrids of this species [17,38]. The *C. reticulata* genotypes were grouped in this cluster with similarity to lemons. However, previous studies have demonstrated a closer genetic relationship with the orange group, as evidenced by similarity indices using SSR [17].

As was previously reported by [33,39], significant genetic variations between *C. reticulata* genotypes have been documented using an SRR array. However, no significant genetic dissimilarity was observed between the six genotypes analysed in the present study (Figure 3). This suggests that other SSRs are required to identify genetic variations within this species.

Considering the intraspecific diversity in the orange group, the lowest variation between the group’s accessions was found between the ‘Laranja Bahia’ and ‘Laranja Pera Mel’ citrus; this low genetic diversity may be due to clonal selection, the presence of cleistogamy, or duplicates in the germplasm [40,41]. In order to observe greater genetic variations in the gene pool among these citrus types, it is necessary to use more markers [15].

The results obtained are an important basis for the adoption of conservation measures, since understanding the genetic diversity of citrus in a germplasm collection can contribute to the control of genetic erosion and the development of citrus breeding strategies, in which the identification and selection of genetically divergent parent plants is essential to broaden the genetic base and enhance breeding efficiency [12].

## 4. Conclusions

This study, using a set of SSR markers as an essential tool for the analysis of genetic diversity, confirmed the existence of significant genetic variability among a citrus population from the germplasm collection of southern Espírito Santo. Furthermore, the markers CCSMEc2 and CCSMEc13 were identified as the most effective for tracking genetic variations in the gene pool at the intra- and interspecific level of *Citrus* spp., considering the species *C. sinensis*, *C. latifolia*, *C. aurantifolia*, and *C. reticulata*.

## Figures and Tables

**Figure 1 cimb-47-00656-f001:**
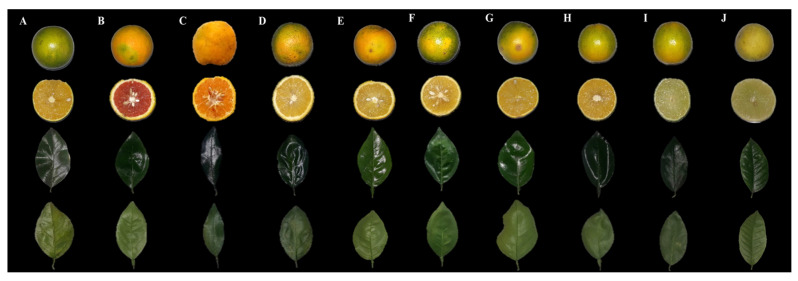
Citrus varieties from the IFES germplasm collection. (**A**) *C. sinensis* (Laranja Pêra Mel); (**B**) *C. sinensis* (Laranja Sanguínea); (**C**) *C. reticulata* (Tangerina Ponkan); (**D**) *C. sinensis* (Laranja Natal Folha Murcha); (**E**) *C. sinensis* (Laranja Bahia); (**F**) *C. sinensis* (Laranja Lima); (**G**) *C. sinensis* (Laranja Seleta Comum); (**H**) *C. sinensis* (Laranja Pêra Rio); (**I**) *C. latifolia* (Limão Taiti) and; (**J**) *C. aurantifolia* (Limão Branco).

**Figure 2 cimb-47-00656-f002:**
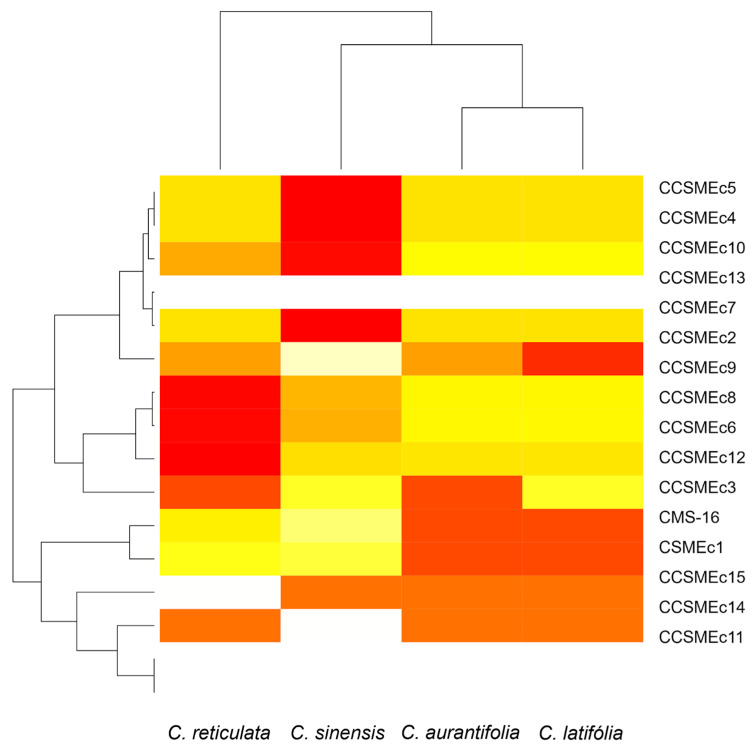
Heatmap with bidimensional hierarchical clustering of the Polymorphic Information Content (PIC) analysis of 16 SSR primers in four *Citrus* species to generate SSR profiles in 60 genotypes. *X*-axis: *Citrus* species. *Y*-axis: SSRs markers.

**Figure 3 cimb-47-00656-f003:**
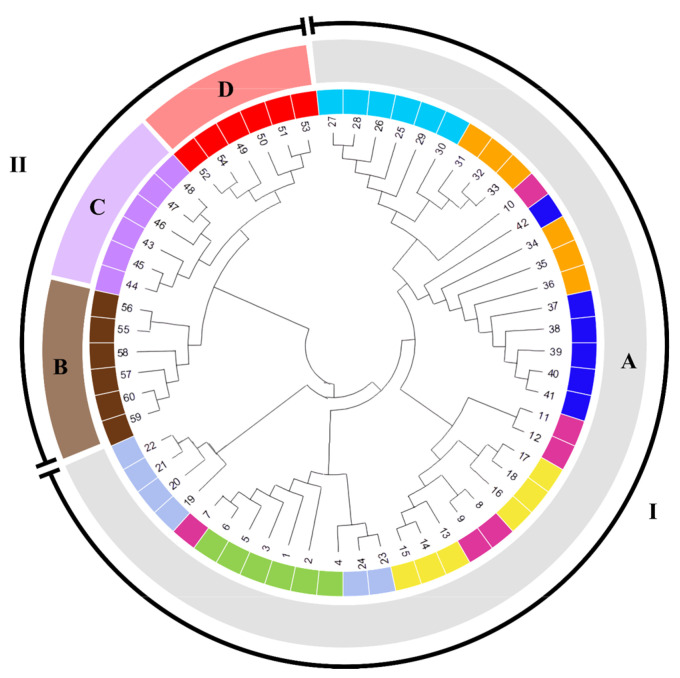
Circular dendrogram showing the genetic similarity between *Citrus* ssp. with Jaccard’s distance using the UPGMA method with a cophenetic correlation of 0.96. The numbers I and II indicate the formation of the clusters. The species are represented by A—*C. sinensis*, B—*C. reticulata*, C—*C. latifolia*, and D—*C. aurantifolia*. The citrus varieties were categorised by colour: sky blue (Laranja Lima); orange (Laranja Seleta Comum); pink (Laranja Sanguínea); dark blue (Laranja Pêra Rio); yellow (Laranja Natal Folha Murcha); light blue (Laranja Bahia); green (Laranja Pêra Mel); brown (Tangerina Ponkan); lilac (Limão Taiti), and red (Limão Branco).

**Table 1 cimb-47-00656-t001:** The name, sequence, fragment size (BP), annealing temperature (°C), and number of alleles of the 16 SSRs for *Citrus* spp.

Name of SSRs	Primer Sequences (5′-3′)	Sequence	Fragment Size (BP)	Number of Alleles
CCSMEc1 F	ACGCTCTCTCCACTATCCGA	(GAA) 10	215	3
CCSMEc1 R	CTGCAGCCGAAGATATGTGA			
CCSMEc2 F	GCTTCTTGGAATGGAGCAAG	(AT) 11	207	3
CCSMEc2 R	CGTTTTTCTGAGGTCACGGT			
CCSMEc3 F	CCATCATGGCTTCTCCAGAT	(TTA) 7	214	3
CCSMEc3 R	TTGCATGTGCCATTGATTCT			
CCSMEc4 F	CTTGCTCGAGTCTACGCTCC	(AG) 14	186	3
CCSMEc4 R	CTTCCTCTTGCGGAGTGTTC			
CCSMEc5 F	ACTGCTGTTCACCCTGTTCC	(CTT) 10	140	2
CCSMEc5 R	GAGAGCTTTCGAGCCTTTGA			
CCSMEc6 F	GCAGCAATTCTGAAGGAAGG	(TAA) 7	158	3
CCSMEc6 R	AGTACAGCATCCTGATCGGC			
CCSMEc7 F	CTTGGAGGAAACAGCAGAGG	(ATC) 8	155	3
CCSMEc7 R	CGAATTGGAATCAAAGGCAT			
CCSMEc8 F	ACCAGAGAGGCTGTGTGCTT	(GAA) 11	161	4
CCSMEc8 R	GTCCACGTAGTCCTTGCCAT			
CCSMEc9 F	TTCGATAGCGCTGTTGTTTG	(GATGAC) 6	280	2
CCSMEc9 R	CACCATCACCATCACGGTAG			
CCSMEc10 F	GGTGGCGAGATTATGCTGTT	(AAC) 7	272	3
CCSMEc10 R	TGCAGTCCCAACAAAAACAA			
CCSMEc11 F	ATCTGCAGGGACAAAACCAG	(GAA) 10 (n) 21 (GAA) 7	213	1
CCSMEc11 R	TCATCTTCACTCACTCGGCA			
CCSMEc12 F	GGAATTCGAGTTGGAGGTCA	(TC) 12	232	2
CCSMEc12 R	ACCACCCATTTGCCTGATAA			
CCSMEc13 F	ATGGCTTTCACAGCATCTCC	(AT) 16	257	2
CCSMEc13 R	TGCATATCCTGAAGACTTTTAT			
CCSMEc14 F	GCCGATCCTCTTTCTCTTTG	(AG) 15ccat (GGC) 7	241	2
CCSMEc14 R	AAGCACGTTATCGGGATCTG			
CCSMEc15 F	TGCCGTTGAGTTTTGATTGA	(GAA) 8	131	2
CCSMEc15 R	GACTGTTGTTCTGATGCCGA			
CMS-16 F	AAAGAAAAATGTTATGTGCATG	(CA) 21	169	4
CMS-16 R	GATGGAGTTTCTCTAGCTCCC			

**Table 2 cimb-47-00656-t002:** Allele frequency and heterozygosity of 60 citrus accessions for 16 SSRs.

Name of SSRs	Allele Frequency				EH	OH
	A1	A2	A3	A4		
CCSMEc1	0.05	0.24	0.75	0.00	0.40	0.10
CCSMEc2	0.44	0.52	0.07	0.00	0.56	0.88
CCSMEc3	0.25	0.16	0.45	0.00	0.65	0.50
CCSMEc4	0.24	0.10	0.66	0.00	0.50	0.68
CCSMEc5	0.45	0.55	0.00	0.00	0.46	0.70
CCSMEc6	0.11	0.30	0.61	0.00	0.53	0.78
CCSMEc7	0.43	0.06	0.55	0.00	0.54	0.90
CCSMEc8	0.05	0.75	0.08	0.10	0.37	0.43
CCSMEc9	0.26	0.74	0.00	0.00	0.38	0.50
CCSMEc10	0.27	0.11	0.61	0.00	0.53	0.78
CCSMEc11	1.00	0.00	0.00	0.00	0.00	0.00
CCSMEc12	0.30	0.70	0.00	0.00	0.42	0.60
CCSMEc13	0.50	0.50	0.00	0.00	0.50	1.00
CCSMEc14	0.30	0.70	0.00	0.00	0.42	0.00
CCSMEc15	0.64	0.40	0.00	0.00	0.46	0.00
CMS-16	0.34	0.42	0.07	0.20	0.68	0.63
Average					0.46	0.53

Expected heterozygosity-EH = 1 − sum (p^2^j); observed heterozygosity-OH = sum (Nij)/[sum (Nii) + sum (Nij)].

## Data Availability

Dataset available on request from the authors.

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
