# Peer review of "Molecular Characterization of Citrus Accessions Grown for Pre-Breeding Purposes"

_cimb, 2025, doi:10.3390/cimb47080656_

Round 1
Reviewer 1 Report
Comments and Suggestions for Authors
This study evaluated the genetic diversity of citrus germplasm in southern Espírito Santo, Brazil, for pre-breeding studies. 16 SSR markers were applied for genotyping 60 citrus genotypes from four species (C. sinensis, C. reticulata, C. latifolia, C. aurantifolia). The key findings were about genetic diversity of the genotypes: 42 alleles were detected, with average heterozygosity of 0.53, and PIC range of 0.21–0.29. The results showed two main groups—Group I (sweet oranges) and Group II (limes and mandarins)—indicating high genetic similarity among sweet oranges. Two SSR markers CCSMEc2 and CCSMEc13 were found the most effective for intra- and interspecific differentiation, identifying high-performance SSRs for future research. This study addresses a gap in molecular characterization of citrus germplasm in Espírito Santo, with practical implications for local breeding programs.
SSR methodology is robust, and results align with prior studies. Logical flow from introduction to discussion, with well-designed experiments and clear visual aids (e.g., dendrogram).
The sample scope limited to four species, more varieties (e.g., grapefruit) would enhance generalizability.
Low-polymorphism SSRs (e.g., CCSMEc11) could be supplemented with SNPs or InDels for higher resolution.
Will the monophyletic origin of sweet oranges be influenced by environmental factors?
Allele frequency tables (Tables 2–3) could be simplified using heatmaps for better readability.
The study is methodologically sound. Integrating phenotypic data (e.g., disease resistance) would strengthen findings.
Author Response
#The sample scope limited to four species, more varieties (e.g., grapefruit) would enhance generalizability.
Response: We appreciate the observation. We acknowledge that increasing the number of species and varieties can contribute to a broader generalization of the results. The species used in the development of this study are those most representative of the region under investigation. However, additional accessions are expected to be incorporated in future stages of the study, through collections in strategic citrus-growing regions of the state and other important producing areas in Brazil.
#Low-polymorphism SSRs (e.g., CCSMEc11) could be supplemented with SNPs or InDels for higher resolution.
Response: We appreciate the suggestion. The accessions used in this study are novel and, until now, had not been evaluated in scientific research. For this reason, we initially chose to use SSR markers. We agree that the inclusion of additional markers, such as SNPs or InDels, can enhance the resolution of the analyses. However, this approach was not adopted in the first phase of the project due to resource limitations. Nonetheless, sequencing-based strategies are planned for the next stages of the study, aiming to complement and deepen the results obtained with SSRs.
#Will the monophyletic origin of sweet oranges be influenced by environmental factors?
Response: The environment can influence the phenotypic expression or regional adaptation of certain cultivars; however, it does not alter the common evolutionary origin that characterizes sweet oranges. This monophyletic origin is considered a historical and genetic trait, resulting from past events of hybridization and selection.
#Allele frequency tables (Tables 2–3) could be simplified using heatmaps for better readability.
Response: We appreciate the suggestion. Table 3, which presents the analysis of Polymorphic Information Content (PIC), was simplified and displayed as a heatmap to improve readability. As for Table 2, since it contains a more detailed analysis, including information on allele frequencies and heterozygosity of the 16 SSR markers, we believe that the tabular format remains the most appropriate to ensure clarity and facilitate data interpretation.
#The study is methodologically sound. Integrating phenotypic data (e.g., disease resistance) would strengthen findings.
Response: This study represents a preliminary stage focused on pre-breeding, aimed at assessing the genetic variability of the accessions. We initially chose to use SSR molecular markers, as this technique is rapid, cost-effective, and highly reproducible. In the future, we intend to integrate phenotypic data, which will allow for more robust analyses with direct applications in genetic improvement.
Reviewer 2 Report
Comments and Suggestions for Authors
Overall, the article is very concise and clear – it sets a clear goal, assessing the genetic diversity of citrus fruits, and this goal is successfully achieved in the course of the study. The relevance of the article for breeding in this regard is undeniable, and thus the article definitely deserves publication. However, there are a number of minor comments that need to be corrected before it can be published.
The use of the Genes program (Cruz, 2016) for the analysis is indicated, but it is not described which functions/algorithms were used to calculate the distance matrix and UPGMA clustering. Please add data.
Throughout the text, citros occurs, probably instead of citros!?
In the title of table 1 - 116SRКs? Probably 16!?
Author Response
#The use of the Genes program (Cruz, 2016) for the analysis is indicated, but it is not described which functions/algorithms were used to calculate the distance matrix and UPGMA clustering. Please add data.
Response:We appreciate the suggestion. This information has been added to the text.
#Throughout the text, citros occurs, probably instead of citros!? In the title of table 1 - 116SRКs? Probably 16!?
Response: We appreciate the suggestion. Correction made.
Round 2
Reviewer 1 Report
Comments and Suggestions for Authors
The manuscript is fine in this edition.